# Application of Droplet Digital PCR Technology in Muscular Dystrophies Research

**DOI:** 10.3390/ijms23094802

**Published:** 2022-04-27

**Authors:** Ioana Lambrescu, Alexandra Popa, Emilia Manole, Laura Cristina Ceafalan, Gisela Gaina

**Affiliations:** 1Laboratory of Cell Biology, Neuroscience and Experimental Myology, Victor Babes National Institute of Pathology, 050096 Bucharest, Romania; ioanalambrescu1130@gmail.com (I.L.); alexandra.gruianu@gmail.com (A.P.); emilia_manole@yahoo.com (E.M.); laura.ceafalan@umfcd.ro (L.C.C.); 2Department of Cell Biology and Histology, Carol Davila University of Medicine and Pharmacy, 050474 Bucharest, Romania; 3Department of Animal Production and Public Health, University of Agronomic Sciences and Veterinary Medicine of Bucharest, 050097 Bucharest, Romania; 4Pathology Department, Colentina Clinical Hospital, 020125 Bucharest, Romania

**Keywords:** muscular dystrophy, droplet digital PCR, absolute quantification, exon skipping, serum biomarkers, cffDNA

## Abstract

Although they are considered rare disorders, muscular dystrophies have a strong impact on people’s health. Increased disease severity with age, frequently accompanied by the loss of ability to walk in some people, and the lack of treatment, have directed the researchers towards the development of more effective therapeutic strategies aimed to improve the quality of life and life expectancy, slow down the progression, and delay the onset or convert a severe phenotype into a milder one. Improved understanding of the complex pathology of these diseases together with the tremendous advances in molecular biology technologies has led to personalized therapeutic procedures. Different approaches that are currently under extensive investigation require more efficient, sensitive, and less invasive methods. Due to its remarkable analytical sensitivity, droplet digital PCR has become a promising tool for accurate measurement of biomarkers that monitor disease progression and quantification of various therapeutic efficiency and can be considered a tool for non-invasive prenatal diagnosis and newborn screening. Here, we summarize the recent applications of droplet digital PCR in muscular dystrophy research and discuss the factors that should be considered to get the best performance with this technology.

## 1. Introduction

Muscular dystrophies are a heterogeneous group of inherited progressive muscle disorders that affects both children and adults. To date, more than 50 types of muscular dystrophies have been described [1]. Among the types of muscular dystrophies, most prevalent in populations and most studied are Duchenne/Becker muscular dystrophy (DMD/BMD), myotonic dystrophy, facioscapulohumeral dystrophy (FSHD), congenital dystrophy, limb-girdle muscular dystrophies (LGMDs), oculopharyngeal muscular dystrophy (OPMD), Emery–Dreifuss muscular dystrophy (EDMD) [1].

Most of these diseases are inherited in an autosomal recessive, autosomal dominant, and X-linked transmission manner. They all share clinical-pathologic similarities in skeletal muscle such as progressive muscle weakness and wasting but differ from each other by various clinical manifestations, the affected muscle groups, age of onset, the rate of progression, and severity of the symptoms [2,3]. Although muscular dystrophies are considered rare diseases, their manifestations are dramatic. In addition to muscle weakness, some muscular dystrophies can even affect the heart muscle, some may result in people needing a wheelchair, and in the case of severe forms such as DMD and congenital muscular dystrophies (CMD), some may even result in early death.

More than 40 years have passed since the first gene and the first protein associated with a form of muscular dystrophy were discovered [4]. A variety of genetic changes (deletions, duplications, insertions, microdeletions, sequence variations, etc.) that occur in different genes encoding for skeletal muscle proteins are responsible for the protein abnormalities [5]. The proteins play a critical role in the proper structure and function of the skeletal muscle and their alterations lead to the occurrence of various types of muscular dystrophies [6].

Over time, research efforts have been directed both toward a better understanding of the pathology of these diseases and to the identification of reliable biomarkers able to provide information on disease progression and regression, find effective and safe treatments, and evaluate various therapeutic approaches, intended to delay the progression of symptoms, thus improving the patient’s quality of life.

Different approaches have also been developed to improve the phenotype, either by restoring protein expression, or by compensating for protein deficiency. Most studies in this regard have been reported in DMD, the disease with childhood-onset and fatal outcome at an early age [7,8,9]. The DMD is caused by a variety of genetic alterations in the *DMD* gene, which encode a protein called dystrophin. Mutations that occur in this gene disturb the reading frame and lead to the absence of dystrophin from muscle. The lack of any treatment for these devastating diseases led to the idea of transforming the fatal phenotype (DMD) into a milder-form (BMD). Several studies have shown that antisense oligonucleotide-mediated exon skipping modulates dystrophin pre-mRNA splicing and restores the open reading frame and thus the production of a truncated but functional protein dystrophin [10]. The CRISPR-associated (Cas) system that induces exon skipping has been also used to correct a specific gene [11]. The skipping efficiencies measurement and copy numbers of transcripts depend on the sensitivity of the molecular techniques used.

The major advances of genetic and molecular biology techniques have made possible the development of specific and sensitive measurement methods and improved diagnosis for each type of muscular dystrophy. The exact knowledge of the mutation, beyond the confirmation of the diagnosis, has a great significance in establishing the carrier status of the disease and also allows patients to benefit from mutation-specific therapeutic approaches under development [12,13].

On the other hand, the discovery of circulating cell-free fetal DNA (cffDNA) in maternal plasma by Lo et al. 1997 [14], opened new directions for research in the prenatal diagnosis field. By assessing the maternal and fetal health as well as the sex of the fetus as early as possible, the birth of children with these diseases might be avoided. Early initiation of therapy in affected patients may also help prevent muscle degeneration [15].

To overcome challenges in all these research directions, the need for a technique that is more sensitive, less invasive, and clearly superior to conventional standard methods such as polymerase chain reaction (PCR) and real-time quantitative PCR (RT-qPCR), is imperative. Droplet digital polymerase chain reaction (ddPCR) has become a more popular method with a growing number of different applications due to its superior sensitivity and specificity.

This article aims to give an overview of the ddPCR technique focusing on its potential utility and applicability for muscular dystrophy research.

## 2. ddPCR Technology

Digital PCR (dPCR) was first described by Sykes et al., in 1992 [16] who evaluated the possibility of diluting template DNA so that, on average, each individual PCR reaction contained only a single template molecule. The method named “limit dilution PCR” or “single-molecule PCR” [17] had the major advantage of individual DNA molecule amplification and a considerably reduced background noise [16]. Subsequently, Vogelstein and Kinzler introduced the term—“digital PCR” for their developed strategy for selective amplification of rare mutation and differentiation of mutant forms from wild-type DNA, in the oncologic field [18].

In the last decade, with the development of instruments, software tools, nanofluidic devices, and emulsion-based formulations, different digitalPCR platforms have been developed. To date, eight companies have offered digitalPCR platforms with different technical characteristics, such as the microfluidic chamber-based (BioMarkTM Digital PCR from Fluidigm), the droplet-based (QX-100/QX-200 ddPCR from Bio-Rad Laboratories and, RainDrop from RainDance Technologies), the micro-well chip-based (QuantStudio3D dPCR from Life Technologies), the microchannels and droplet-based crystal digital PCR (Stilla Technologies Naica), the nanoplate-based system (QIAcuity from Qiagen), the chip-in-a-tube technology (Clarity™ from JN MedSys Clarity) and semiconductor chip-based (LOAA dPCR from Optolane). Technological advances in microfluidics continue to emerge, offering new and creative solutions, thus improving the performance of current digital platforms.

The growing number of studies in which this technology is used highlights its impacts in many areas of health sciences and medical research.

Among the most available commercial platforms up to now, the droplet-based system has been the most used in the different muscular dystrophies research [19,20].

In general, a schematic workflow of ddPCR technology includes the steps shown in Figure 1.

Overall, the ddPCR technique is based on partitioning the nucleic acid samples with a certain concentration into approximately 20,000 nanoliter-sized droplets using a water-oil emulsion system, PCR amplification of each droplet, and individual droplet analysis based on the fluorescent signal [21]. As a result of the partitioning of the DNA template into thousands of nano-liter droplets, each droplet can contain either one or zero copies of the target molecules. Ideal droplets contain only one or no target DNA molecule. During PCR cycling, only positive droplets (with at least one copy of target DNA molecule) emit increased fluorescence and the value of “1” is assigned to each positive droplet while a negative droplet is assigned a value of “0”. The cases with two or more targets per droplet would hamper the analysis and should be eliminated [22].

The detection of droplets with positive or negative targets is achieved with the help of a two-color detection reader for different fluorophores such as fluorescein amidine (FAM) and hexachloro fluorescein (HEX) or Aequorea Victoria (VIC) [21,22,23]. The positive droplets are checked and counted in all reactions. Positive droplets are considered to be all those that have a higher fluorescence intensity than the set threshold, while negative droplets are all those with fluorescent intensity below the threshold. Droplet counting is done automatically by the software and provides absolute initial quantification of the targets expressed as copies per microliter. The precise quantification of droplets with target molecules or the estimation of the mean number of copies per partition involves the use of Poisson statistics [24,25].

## 3. Advantages and Disadvantages of ddPCR

The PCR has undergone changes over the past decades to improve sensitivity and specificity in the context of amplifying and detecting nucleic acids in low concentrations. Due to cumbersome maneuvers and a low limit of detection, the visualization of PCR products on gel electrophoresis has been gradually replaced in many laboratories with the RT-qPCR method.

The third generation of PCR—the digital PCR—allows for a more accurate and quantitative analysis of the initial target, which makes it a valuable tool for molecular detection [26]. Accuracy and sensitivity are two extremely important goals in molecular diagnosis. While qPCR may not excel in situations that require the detection of a weak mutant signal surrounded by genomic noise (caused by wild-type sequences), the ddPCR method is demonstrably superior in various applications. Thus, ddPCR technology has emerged as a promising molecular technique proficient for quantifying target sequences and identifying rare variants [27,28].

The advantage that results from this method consists of the concentration of the target sequence. In this way, one can expect a reduction in template competition, which allows the detection of a weak mutant signal in a wild type of background. Moreover, the target concentration due to partitioning could also enable the ddPCR reaction to be more resistant to inhibitors [29]. Another advantage of the way the sample is dispersed into individual reaction chambers is that of a decrease in the degree of cross-contamination between neighboring compartments [21]. In other words, the difference between the ddPCR and the standard quantitative PCR techniques relates to comparing absolute to relative target quantification [30].

The ddPCR technique can achieve a sensitivity of up to 0.1% (in certain situations even up to 0.001%), contrasting with 1% for the traditional PCR method [31].

Reproducibility is another improvement of the ddPCR technique, especially since qPCR is more prone to contamination, which may result in the inhibition of the Tag polymerase or influence on primer annealing [32,33].

The clinical practice needs a robust and accurate method of quantification for different targets and the ddPCR technique meets these requirements in terms of absolute quantification. The random nature of target distribution together with Poisson statistics overcome the use of standard curves, delivering a result expressed as copies of target per microliter of reaction [34]. An overview of the similarities and differences between the two PCR techniques is presented in Table 1.

Compared to traditional PCR, the RT-qPCR method has the advantage of collecting the data in the exponential growth phase with an increased dynamic range of detection [31]. In addition, the RT-qPCR method offers the opportunity of running parallel reactions, thus minimizing potential errors [35]. However, the method is prone to significant biases due to a certain sensitivity to inhibitors, which results in different amplification efficiencies [36].

## 4. Hallmarks of a Well-Designed ddPCR Assay

The design assays are similar to those for qPCR. As with all PCR generation, the success of ddPCR use depends on well-designed and optimized assays. Poor design, as well as non-optimal PCR conditions, can lead to nonspecific amplification and ambiguous results. To achieve the best performance with the ddPCR technique, a robust optimization of the parameters and reaction conditions is required. Among these key factors [37] that influence the reliability of digital PCR measurements, several must be mentioned:(1)Nucleic acid samples concentration. As well as other molecular biology techniques, the quality, and the quantity of nucleic acid samples, may affect the result and are essential for the accuracy of the assay. No special requirements are necessary regarding the sample preparation. However, it should be noted that some methods of nucleic acid isolation may interfere with the generation of droplets [38] and, therefore, the method must be chosen which offers a good separation of positive and negative droplets as well as the best signal intensity. Qubit (Invitrogen, Waltham, USA) and Nanodrop (Thermo Scientific, Waltham, USA) measurement of nucleic acid samples concentration and purity is essential to achieve reliable results [39].

For optimal results, the input amount of nucleic acid introduced into a reaction must be optimized. The partitioning step in ddPCR is important to digital assays and presents some advantages such as (i) better precision when counting copies of targets (it measures individual molecules); (ii) the enrichment effect increases the ratio of the target of interest and raises amplification efficiency of low-abundance mutant nucleotides against wild-type background and even the tolerance to inhibitors. In the partitioning step, the surfactants must be optimized to avoid interfering with the assay. Additionally, in the ddPCR technique, the errors from partitioning appear due to the different distribution of targets among partitions from one experiment to another [40].

(2)The design of primers and probes are among the most critical factors for the success of the experiment and should be carefully done to avoid self-annealing or cross-reactivity. Whether or not a design program is used, for primers and probes, the same rules as for qPCR analysis must be apply.(3)Assay optimization. Achieving accurate interpretable results requires a number of important factors to be considered when optimizing a ddPCR assay. The annealing temperature must be optimized for each target using a gradient PCR range between 55 and 65 °C, an interval in which most targets have an optimal temperature. A temperature is optimized when the largest separation between positive and negative droplets is achieved [37,41].

Several reports have shown that the concentrations of primers and probes in a ddPCR reaction influence the fluorescence amplitude of the measured droplets [42], so a higher concentration leads to an increase of amplitude, thus allowing better separation of specific signals from background noise. The best results were reported with a final concentration of primer of 0.9 μM and 0.25 μM for the probes [43]. In multiplexed digital PCR assays, the detection of target sequences can be influenced by the intensity or concentration of fluorescence-labeled probes [44]. For instance, the quantification by the ddPCR technique considers that partitions have identical volumes, but sometimes a degree of variability in volume can be observed. The precision of the ddPCR method is limited also by the specimen sampling, whose effect is prevalent for low target concentrations and is reduced by using replicates [45]. Samples analyzed in duplicate, or triplicate, may prevent bias in quantification due to pipetting errors. Summing the data from duplicates increases the number of measured events.

(4)Controls. In ddPCR technology, the use of a reference gene is not mandatory because of the absolute quantification of the number of targets from a sample. Furthermore, the assays can be affected by technical problems associated with the reverse transcription step. Primer dimers and secondary structures are avoided, and the annealing temperature can be used for reaction optimization. An important aspect of the ddPCR assay is represented by the appropriate use of a specific set of controls that are important for method performance [46,47]:
(i)negative controls—for monitoring a false-positive reaction, which may be a marker of contamination or a poor design of primers/probes, and for the determination of limit of detection (LoD);(ii)positive controls—useful to test for whether the template amplification occurs under the established reaction conditions;(iii)non-template controls (NTCs)—for control of contamination in all reagents [48]. Poor design optimization can lead to a bad assay performance.

## 5. Applications of ddPCR in Muscular Dystrophy Research

So far, many publications have reported the application of the ddPCR technique in various fields, such as diagnosis of infectious disease [49,50], pathogen detection [51,52], human biomarker screening [53,54,55], gene expression [56], identification of species [57], and food and environment monitorization [58], etc.

Most of the applications related to the ddPCR technology originate from its high-precision absolute quantification. In the last few years, a growing number of studies have reported an increased interest in the applicability of ddPCR methods in many areas, including muscular dystrophies, where precise identification is required. The method has proved to be useful for molecular analysis in determining the efficacy of various therapeutic approaches, miRNA biomarkers analysis, copy number variation, and prenatal diagnosis.

### 5.1. Absolute Quantification

Among the most used applications of ddPCR technology is the absolute quantification (ABS). This refers to an experimental design used for the quantification of target sequences and nucleic acids in copies per microliter for a given sample.

The majority of the research reported for muscular dystrophies using this application has been for the precise quantification of nucleic acids [54], to validate various drugs efficiency [43], and for biomarker analysis [54,55].

Many promising therapeutic approaches for these diseases are currently under investigation, several of them being tested in clinical trials [59,60,61].

The DMD, the most severe and common disease form, is also the most studied. The mutations that occur in the *DMD* gene cause the two phenotypes: the DMD and the milder one BMD, depending on whether the mutation disrupts the reading frame or not [62]. Mutations that disrupt the open reading frame led to the complete absence of dystrophin from skeletal muscle, resulting in the severe DMD phenotype. The mutations that maintain the reading frame allow the synthesis of a truncated but functional protein underlying a milder phenotype of BMD. The lack of treatment for DMD patients suggested the idea of transforming a severe phenotype into a less severe one [10] by the restoration of the *DMD* gene open reading frame [63].

The recent emergence of genome-editing technologies through which mutations that cause the disease can be corrected became an attractive option for therapeutic applications in muscular dystrophies. These therapeutics include antisense oligonucleotides (ASOs), target RNA molecules, and CRISPR/Cas9, a promising genome editing tool that directly targets genomic DNA.

The ASOs mediated exon skipping is one of the most promising strategies for DMD that has been developed to induce the omission of one or more exons, thus allowing the restoration of the reading frame and the synthesis of a shorter protein [10]. Previous reports have shown that restoration of around 15% of normal levels of dystrophin is sufficient to protect muscle from contraction-induced injury [64] and 30% to avoid muscular dystrophy in humans [65]. This approach was done using single- or double-stranded pieces of modified nucleic acids, ASOs that have a complementary sequence to the pre-mRNA, masking one or more exons to be skipped. Thus, during the protein production, the masked exons are ignored leading to restoration of the reading frame [10,66,67].

The exon skipping strategy was first developed as a possible treatment using the *mdx* mouse model that carries a spontaneous nonsense mutation in exon 23 of the *DMD* gene [68]. The recently created del52hDMD/*mdx* mouse model, genetically and functionally characterized, which carries both murine (with stop mutation in exon 23) and human (with deletion of exon 52) *DMD* genes, exhibits muscular dystrophy and shows impaired muscle function [69]. It enables testing the effects of treatment with specific ASOs targeting human exon 51 or exon 53 on RNA, on protein level, and showing the degree of histological and functional recovery. Hiller et al., [70] reported the most accurate quantitative method results using the ddPCR method [19] for absolute quantification of exon 51 skipping levels in del52hDMD/*mdx* mice.

The most studied ASOs for exon skipping are 2′-O-methyl-phosphorothioate (2OMePS) [71] and phosphorodiamidate morpholino oligomers (PMO) [72]. Quantification of induced exon skipping levels is critical both for ASOs selection and for assessing drug therapy by comparing transcript levels before and after ASOs treatment.

For absolute quantification of exon skipping levels, the ddPCR technique has provided the most accurate results. Compared to previous PCR-based approaches, ddPCR technology was able to measure a very low amount of exon 51 skipping at baseline in a DMD muscle cell culture [19]. Currently, four exon-skipping therapies are approved for DMD patients amenable to different skipped exons. Approved drugs designed for a specific mutation in the *DMD* gene (Eteplirsen-exon 51 [73], Golodirsen-exon 53 [74,75], Viltolarsen-exon 53 [76], and Casimersen-exon 45 [77,78]) restore the reading frame and produce a shorter but functional protein contributing to amelioration of the phenotype.

A multicenter comparison of ASOs-induced quantification methods for ignoring exon 51 in patients with DMD showed that the ddPCR technique is the most precise method for assessing the level of exon omission without overestimating the quantification result [79].

Because these treatments have succeeded in restoring only a variable level of protein, Novak et al., [80] evaluated the turnover dynamics of restored dystrophin and dystrophin-glycoprotein complex (DGC) proteins in mdx mice after exon skipping therapy. The study established the efficacy of a single bout of exon skipping therapy in *mdx* mice. By use of ddPCR, the assessment of mRNA transcript stability declines and skipped mRNA levels was confirmed.

For premature stop codon in the *DMD* gene induced by nonsense mutations, Ataluren (Translarna, PTC124), was designed as a medication that induces readthrough of premature stop codons during mRNA translation [81,82], leading to restoration of full-length dystrophin protein. However, the Ataluren efficacy for DMD patients has not yet been well elucidated. Several studies have been conducted to understand the mechanism of action as well as efficacy data on the animal models [43] and DMD non-ambulant patients [81]. To assess whether a significant improvement in muscle strength was achieved after treatment with Ataluren in a dystrophin-deficient zebrafish mutant line dmd^ta222a^, the level of dystrophin transcript was quantified. By ddPCR methods, the beneficial effect of Ataluren treatment over 5 days was confirmed, showing that the method is a reliable analytical technology for sequence-specific detection and precise quantification [43].

The ddPCR technique has continued to develop over the last 20 years, moving from the in vitro to in vivo models and currently has reached the stage of clinical trials with encouraging results. Due to the accuracy, high sensitivity, and reproducibility of the results, ddPCR has been proposed as a powerful technique for the precise absolute quantification of exon skip efficiencies of ASOs in (pre)clinical development for DMD therapeutic studies [19].

The CRISPR-Cas9 genome-editing strategy is also used as an effective method to alter the genome [83], offering accurate results in muscular dystrophy research. For DMD∆52 myoblast cultures model edited with CRISPR/Cas9 [84], dystrophin expression was assessed with accurate results by the ddPCR technique for confirmation of the abolishment of the exon 52 from the *DMD* gene, compared to control myotubes.

For limb-girdle muscular dystrophy type 2A (LGMD2A), a progressive muscle disease caused by genetic defects in the *CAPN3* gene, several potential therapies have been developed. Among these, adeno-associated viral-mediated therapy (AAV) is the current standard tool for gene transfer in skeletal muscle. A recent study [85] shows efficient *CAPN3* transgene expression in muscle tissues of a murine model, by RT-ddPCR, after local injection with rAAV vectors expressing *CAPN3.* The biodistribution of rAAV9 was also assessed by ddPCR methods using DNA extracted from major organs and different types of skeletal muscles.

### 5.2. Copy Number Variation

Copy number variation (CNV) is described as a structural variation comprising DNA fragments greater than 1 Kb in size [86]. This phenomenon causes a repetition of different fragments of the genome and represents an important source of genetic variation. Consequently, CNV can encompass duplications or insertions, deletions, or even the rearrangement of the genome [87]. Moreover, it is possible that within the same species the copy number of genomic segments could present an inter-individual variability [88]. Some CNV are inherited, however, others arise de novo [89]. It has long been recognized that CNV is an important risk factor for several diseases, due to their implication in human physiological functions [90]. In terms of consequence, CNV can influence multiple protein-coding genes and regulatory regions, which ultimately will affect cell physiology at multiple levels [91]. A possible explanation for disease causation and for most of the undiagnosed cases of inherited muscular disorders could be represented by missed CNV. Additionally, Piluso and colleagues estimated in a study comprising the analysis of 245 genes involved in neuromuscular disorders that the frequency of CNV is between 4 and 10% [92].

The accurate quantification of mutant RNA molecules is also important for helping to understand the diseases’ complexity as well as for efficacy assessment of various therapeutic drugs. A study regarding myotonic dystrophy type1 (DM1) and type2 (DM2) [93], showed that in DM1 there is an abnormal expansion of CTG repeats in the 3′-UTR of the *DMPK* gene, and in DM2 it was observed the existence of elongated CCTG repeats in intron 1 of *ZNF1/CNBP* gene. The ddPCR and MLPA (Multiplex Ligation-dependent Probe Amplification) techniques were used by Wojciechowska et al. [93] to study the mutant DMPK transcript (DMPKexpRNA) and the aberrant alternative splicing in DM1 and DM2 in human tissues and cells. By using ddPCR methods to calculate the absolute number of DMPK transcripts in copies per cell, the study offers a new quantitative approach that highlights the benefits of the absolute quantification method without external references [93].

Other RNA analyses using ddPCR techniques have revealed that extracellular RNA (exRNA) splice products in human urine may be utilized as biomarkers for two forms of muscular dystrophies, DM and DMD [54]. Urine from patients with DM type 1 contains ten transcripts that are spliced differently in exRNA. In DMD patients treated with the antisense oligonucleotide drug (Eteplirsen), detection of mutation-specific DMD mRNAs in urine, constitute confirmation of exon-skipping activity of the drug.

### 5.3. Gene Expression and miRNA Quantification

MicroRNAs (miRNAs) are small, non-coding RNAs (19–22 nucleotides long) regulators of gene expression that can contribute as promising biomarkers for early diagnosis, diseases prognosis, as well as the assessment of therapy efficacy.

In skeletal muscle, it has been shown that miRNA plays an important role in the development and function of skeletal muscle [94]. At the same time, the altered expression of several miRNAs plays an important role in skeletal muscle pathology including muscular dystrophies.

A series of potential biomarkers were recorded in DMD patients and animal models, including *mdx* mice, dystrophin/utrophin double-knockout (dKO) mice, golden retriever muscular dystrophy (GRMD), and canine X-linked MD in Japan dogs (CXMDJ).

Five miRNAs were found in the serum of DMD patients: miR-1, miR-133a, miR-133b, miR-31, and miR-206, expressed in muscles, and called dystromirs [94,95,96]. The expression of miR-1 is initiated by myogenic regulatory factors and promotes terminal differentiation. The miR-133a increases myoblast proliferation and is expressed from the same transcript as miR-1. mir-206 together with miR-133b are encoded by a single noncoding RNA from skeletal muscles [95]. The miR-206 is also expressed in satellite cells, in muscle precursor cells, and promotes their differentiation and fusion into multinucleated myotubes and mediates the increase of utrophin expression in skeletal muscles [97].

The serum miRNAs were analyzed also in mouse models for muscle pathologies such as DMD (*mdx*), limb-girdle muscular dystrophy type 2D (LGMD2D; sgca-null mice) and type 2C (LGMD2C; sgcg-null mice) and EDMD. Vignier et al. [98] showed that the upregulation of miR-1, miR-133a and miR-133b in serum was detected in the mouse models for DMD, LGMD2D, LGMG2C but the three miRNAs were slightly downregulated in the model for EDMD. The study of serum/plasma miRNAs in muscular dystrophies involves the use of RT-qPCR using a standard curve for absolute quantification of endogenous or exogenous spike-in miRNAs [93,99]. Thus, the widespread use of miRNAs as biomarkers in muscular dystrophies calls for the development of methodologies capable of detecting miRNAs accurately and reproducibly in biological fluids [100]. However, non-invasive identification of biomarkers for muscular dystrophies, to exploit their potential require the improvement and standardization of their detection and quantification methods.

Llano-Diez et al. [101] used ddPCR techniques in a study for quantification of the copy number of circulating miR-30c and miR-181a in serum from DMD and BMD patients. The authors found that miR-30c and miR-181a were both elevated and highlighted that circulating levels of these two dystromirs had a sensitive and specific diagnostic value. They also demonstrated for the first time the value of ddPCR as a powerful technique for the quantification of miRNA in muscular dystrophy.

The study performed by Trifunov et al. [102] regarding the role of miR-181a-5p, miR-30c-5p, and miR-206 as prognostic biomarkers for long-term follow-up of DMD and BMD patients also used the ddPCR technique. The authors showed that the levels of miR-30c and miR-206 remained elevated in DMD patients compared to controls, but also, that miR-206 can successfully discriminate the DMD and BMD phenotype, regardless of the disease stage. In the same study, the longitudinal analysis showed that the miR-181a levels were high, both in DMD and BMD patients, but there was no significant difference observed in the second and third time points. Additionally, the authors noted that miR-206 levels in BMD patients are intermediate between those in DMD patients and controls and that although miR-206 levels drop with age, they remain higher in DMD patients. Interestingly the study showed that miR-30c has the ability to distinguish the DMD population from healthy subjects [102]. Another important observation of the study was the difference in miR-206 level determined by ddPCR techniques in very young patients with DMD, compared to another study in which the level of the same miRNA was quantified by qPCR. This difference can also be attributed to the sensitivity of the two methods.

Several studies suggest that ddPCR techniques can be a powerful tool in the accurate quantification of miRNAs as biomarkers for DMD and BMD disorders, for monitoring the progression of the diseases, or even for patients’ response to therapy [103]. Elevated levels of these microRNA in the serum of DMD/BMD patients were reported to be 6–7 times higher than normal. Consequently, they proposed these myomiRs (muscular microRNAs) as biomarkers for diagnosis and assessment of disease severity and the use of ddPCR techniques as an accurate method to quantify small amounts of nucleic acids [94,96].

For FSHD, a type of muscular dystrophy caused by de-repression of the *DUX4* gene, the inhibition of this gene could be a therapeutical approach, employing RNA interference (RNAi). Saad et al. [104] reported a strategy to inhibit DUX4 directing RNAi against the gene using a natural microRNA, miR-675, in cellular FSHD models, at the same time protecting the muscles from DUX4-associated death in mice. The ddPCR technique along with qPCR were used to confirm that miR-675 significantly reduced DUX4 mRNA expression.

In recent years, significant progress has been made in both characterizations of the genetic defects leading to muscular dystrophies and the associated histological features. However, the pathology of these diseases is poorly understood. Over time, a series of experiments have been performed to add new insights into the molecular pathology of these diseases [105]. Comparison of gene expression between an affected and unaffected muscle in animal models [106,107] and DMD patients [108,109,110] was performed recently with flow cytometry or qPCR technology.

Even though the main use of ddPCR is to detect CNVs and rare mutations in genomic DNA, it is also a useful technique for gene expression analysis. A growing number of articles report the use of ddPCR technology in gene expression studies.

The ddPCR technique was used for confirmation of flow cytometry results in a comprehensive approach that aims to improve understanding of innate immunity in dystrophic muscles in order to develop specific anti-inflammatory treatments [111]. In the *mdx* and Het mouse models, the muscle cytokine protein levels and cytokine receptor gene expression levels were compared together with flow cytometric analysis of immune cell populations. The ddPCR technique was used to corroborate flow cytometry findings on a few targets (Mrc1, Spp1, Nos2, Col1a, and Timd4) from four sorted populations from mdx and Het quadriceps [111].

Two new cell culture models created by CRISPR/Cas9 gene editing have been developed for use in the preclinical evaluation of new DMD therapies [84]. One cell culture model replicates a patient’s deletion (DMDΔ52-Model) which causes no dystrophin synthesis, and the other one overexpresses utrophin (DMD-UTRN-Model). In addition to immunohistochemistry, western blot and ddPCR technology were employed to validate models displaying dystrophin inhibition and utrophin overexpression. The expression of myogenic factors (Myf5 and MyH3) was analyzed at different fusion times using ddPCR techniques, both in DMD∆52-Model cell cultures compared to control myotubes, and in DMD-UTRN-Model cultures compared to DMD myotubes.

### 5.4. Non-Invasive Prenatal Diagnosis

The non-invasive prenatal diagnosis (NIPD) started after the discovery of cell-free fetal DNA (cffDNA) in maternal plasma by Lo et al. [14], thus avoiding invasive procedures such as amniocentesis or chorionic villus sampling. The health of pregnant women represents an important public health issue. There is a general tendency across developed countries toward advanced maternal age. This fact has implications on the health of the fetus predisposing it to various fetal abnormalities. Prenatal genetic testing has evolved considerably over the past decades, and new tests have been introduced into the prenatal setting at a rapid pace [14]. A major breakthrough for the investigation of prenatal genetic defects came when the PCR technique was developed.

For sex-related diseases, fetal sex determination at the early gestational period is important for the prediction of pathological phenotype in an unborn child, especially in families at risk of X-linked disorders, such as DMD, BMD, and EDMD types [112]. In 2018, D’Aversa et al. [112] reported that the qPCR testing is not trustworthy and accurate for fetal sex diagnosis when performed using blood withdrawn at the early gestational period, in particular prior to 7 weeks of gestation due to the low amount of cffDNA relative to circulating maternal DNA. Other groups reported that the difficulties of qPCR-based methods to identify male fetal sex, as well as different pathogenic variants at early gestational stages, are due to low amplification, generating an unreliable quantification or false-negative results. However, the problems described seem to be due to the insufficient amount of circulating fetal DNA present in the maternal plasma in the first weeks of gestation (12 weeks or less) [113] and the reduced sensitivity of the method. Determination of fetal sex from maternal plasma at early gestational stages (4.5 weeks) by ddPCR techniques reveals the superiority and accuracy of this technique when compared with qPCR.

All these results demonstrate that ddPCR is a powerful technique with great potential in prenatal diagnosis in the early gestation period. The methodology has proven useful for the evaluation of low-level mosaicism providing insights into germline mosaicism. Lately, there has been reported the identification of somatic mosaicism in the *DMD* gene. The ddPCR analysis has shown to be more precise for the quantification of mutant alleles than estimations based on electropherograms or quantitative PCR analysis [114].

Jin et al. [115] recently showed that ddPCR technology may identify low level germline mosaicism in couples with no clinical manifestations of dystrophinopathy. They studied a couple with two consecutive conceptions with the same deletions in exon 51 of the dystrophin gene [115]. The mutant frequency in the mother was 3.53%, which is a low-level mosaicism unidentifiable by the MLPA technique, for example. Thus, ddPCR has proved a very good tool for the evaluation of de novo mutations recurrent risks.

## 6. Conclusions

Muscular dystrophies are being evaluated differently as we move into the new causative genetic age. Due to the current lack of curative treatment, the genotypic spectrum of most syndromes and the phenotypic diversity of individual genes will undoubtedly define the future of these illnesses. In this context, new techniques of molecular biology have been developed such as the ddPCR technique that has been used in a variety of clinical states. An increasing number of publications show that the ddPCR technology is gaining new value in both research and diagnosis. Based on the experimental results in which ddPCR technology was used, it appears that the most suitable applications are those in which precise and accurate quantification is required, providing valuable information about the response to different genetic therapies that aim to remove the genetic defect.

The method is also able to detect a very low level of spontaneous exon skipping level before and after ASOs treatment, which is important in the drug efficacy evaluation.

Specific molecular biomarkers have provided valuable information about disease progression, response to treatment, and unaffected muscle mass ratio. Their investigation using ddPCR technology may provide a less invasive method of monitoring compared to muscle biopsies. However, further studies are needed to validate them for clinical practice.

Furthermore, ddPCR technology allows early detection of the risk of X-linked pregnancies, and prevents the birth of affected children, while early management of the affected patients could determine the initiation of therapy before the onset of the first symptoms. Due to the increased sensitivity, ddPCR techniques can be also a non-invasive prenatal diagnostic method by cffDNA analysis, being an alternative to traditional invasive prenatal diagnosis methods such as chorionic villus sampling and amniocentesis for the various diseases at risk.

For all these reasons, ddPCR techniques could help improve the diagnosis and clinical management of many other diseases and not only for muscular dystrophies. Compared to other available methods, ddPCR has obvious superiority in terms of sensitivity, specificity, a low limit of detection, and highly reproducible results. However, this methodology still has some limitations regarding the false-negative as well as false-positive results and experimental artifacts, but the continued development of this technology could help in improving the management of the various diseases.

## Figures and Tables

**Figure 1 ijms-23-04802-f001:**
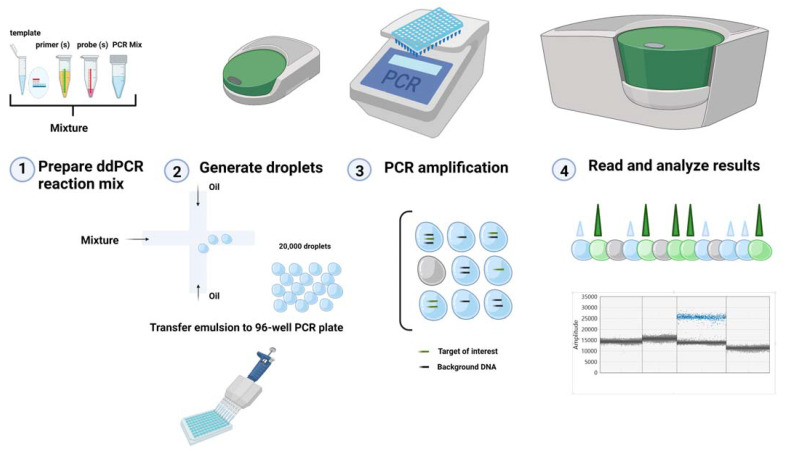
Schematic illustration of ddPCR (Created with Bio.Render.com). This figure illustrates a typical ddPCR workflow. (**1**) For the assay, a quantity of ddPCR mix, (buffer, dNTPs, primers, and probes) and the DNA sample are loaded in a multi-channel cartridge with droplet generation mineral oil. (**2**) The droplet generator creates a vacuum with negative pressure crosswise the cartridge. In this way, the negative pressure subdivides the DNA sample into water-in-oil droplets at the same time. (**3**) Subsequently, these partitions will be individually amplified under specific thermal cycling conditions. (**4**) After PCR amplification is complete, the plate is placed in the droplet reader, which analyses each droplet individually.

**Table 1 ijms-23-04802-t001:** The similarities and differences between ddPCR and RT-qPCR.

Strengths		Similarities	Differences	
qPCR	ddPCR	ddPCR/qPCR	qPCR	ddPCR
Gold standard technique for target DNA quantitation and gene expression analysis	High precision quantification at low input copy number sequence in a complex background	Both methods have multiplex capability	Relative measurement	Absolute measurement
Economic costs	High sensitivity	Both methods are easy to use.	Standard curves needed	No need for calibration or standard curves
Rapid test results	Independent analysis and data processing of samples	Quantification of the amount of target in a certain sample	No sample partitioning	The sample is partitioned into a large number of individual reactions
	High tolerance to PCR inhibitor	The same components used in the reaction (PCR Master Mix, primers, fluorescent probes (Taqman probs FAM and HEX/VIC)	Real time PCR data acquisition	End point data collection

## Data Availability

Not applicable.

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
