# Peer review of "Application of Droplet Digital PCR Technology in Muscular Dystrophies Research"

_ijms, 2022, doi:10.3390/ijms23094802_

Round 1

Reviewer 1 Report

The REVIEW manuscript entitled “Application of droplet digital PCR technology in muscular dystrophies research.” (Manuscript ID:  ijms-1640771) by Dr. Kogoji and colleagues provides a detailed description of the utility and applicability of the current, non-invasive, droplet digital PCR approaches for the muscular dystrophy research as well as clinical application. The review is interestinga nd concise. Despite several improvements are necessary to be made, the manuscript is relatively well written. The scientific writing style is adequate. Figures should be improved in the quality. In general, the ms will improve our knowledge on droplet digital PCR and clinical applications for Muscular dystrophy research, management and diagnosis/prognosis. I therefore recommend a minor revision

I have several observations for improving the work: 

General comments
1.    A relatively large number if nations/sentences are lacking in supporting references. For instance lines 144-149, 159-165, 439-445 and others
2.    For a better reading, and considering the general information included in the “5. Advantages and disadvantages of ddPCR” section, I suggest moving the aforementioned section after the “2. Droplet digital polymerase chain reaction (ddPCR) technology” section
3.    The quality of Figure 1 should be improved. Moreover, more details on the workflow should be included in the caption
4.    The notion that the ddPCR gives an absolute quantification of the target molecule, compared to the relative quantification given by standard quantitative PCR, should be underlined in the text. Authors can include (DOI 10.1186/s12575-021-00144-w)
5.    If present, studies reporting dysregulated miRNAs in muscle tissues from DMD patients should be included in section “4.4 Gene expression”

Minor 
Line 15 bold style should be removed
Line 24 “droplet digital PCR” it should be “droplet-digital PCR (ddPCR)”
Line 203 ddPCR has also been implemented for detecting viral infections (DOI 10.3389/fmicb.2020.591452). This important information/reference should be included
Line 220 “app” application? Please revise 
Line 224 clinical trials names/ IDs?
Lines 225-231 please include supporting references
Line 268 please remove the double parentheses
Line 292 better ddPCR
Line 445 “results, However” a period should be included

Reviewer 2 Report

In the review (ijms-1640771), Lambrescu et al. attempted to summarize the applicability of droplet digital PCR (ddPCR) in the muscular dystrophy research field.

After a bibliographic search, the authors stated that the droplet-based QX-200 ddPCR system from Bio-Rad Laboratories is the most used system for the study of muscular dystrophies. This statement may change in the future as new digital PCR systems with new capabilities have been developed in the past couple of years and as they could also become important in the field, they should be mentioned and that would improve the quality of the review.

The authors failed to compile the concepts and studies included in the review and consequently multiple ideas and sentences are repeated throughout the text, as well as inconsistent abbreviatures and wrongly formulated sentences. The authors should revise the content of the review to improve the flow and readability of the text.

Round 2

Reviewer 2 Report

After the revision, authors have addressed some of the comments, but the manuscript still requires some changes.

Line 28: …performance with the this technology

Line 38: close parentheses after (FSHD, congenital dystrophy

Line 45-47:

“In addition, to muscle weakness, some of them can even affect the heart muscle, people may need a wheelchair and, in the case of DMD, even death may occur.”

This sentence needs to be reformulated. In the present form, reader may understand that only DMD is fatal, and that statement is false, other congenital muscular dystrophies also have shorter lifespan.

Line 68 and 291: Do you mean milder instead of “middle”?

Line 82-83:

“By assessing the maternal and fetal health as well as the sex of the fetus as early as possible, the birth of children with these diseases is avoided.”

Perhaps better to say, “might be avoided”.

Figure 1: There are two numbers 1 and there is a typo in step 4, analyze.

Line 301-302:

“The most widely reported methods as genome-editing strategies are based on the use of antisense oligonucleotides (ASOs) or CRISPR-Cas9”.

This sentence might create confusion to the reader. Genome-editing strategies are strategies aiming to edit the DNA. By contrast, antisense oligonucleotides (ASOs) target RNA.

Line 305: Previously reports have…

Line 340-341:

“…in mdx mice that harbors a nonsense point mutation in exon 23 in the DMD gene.”  

This has already been mentioned in line 313.

Line 402-403:

“DMD patients have mutation-specific DMD mRNAs in urine, a confirmation of exon-skipping activity of antisense oligonucleotide drug eteplirsen.”

This sentence needs to be reformulated. I do not see why the presence of mutation-specific DMD mRNAs in urine is a confirmation of exon-skipping activity of antisense oligonucleotide drug eteplirsen.

Line 414-417:

“Biomarkers are indicators of different pathogenic conditions or pharmacological responses to therapies (60). Furthermore, they are at the same time, extracellular biomarkers for different muscle diseases including DMD.”

This information was already given in a more general way in the previous sentence. Repeated information should be avoided.

Line 498: Please check the following sentence:

Two new cell culture models created by editing the CRISPR / Cas9 genes have been developed for use in the preclinical evaluation of new DMD therapies”  

Editing the CRISPR/Cas9 genes?

Line 559-561: Please check the following sentence:

“Further studies are required for their validation However, further studies are needed to validate them for clinical practice”

Line 569-571:

“Compared to other available methods, ddPCR has obvious superiority in terms of sensitivity, specificity, reproducibility, a low limit of detection, and highly reproducible results”

Repeated information in the same sentence.
